# Antibody–Drug Conjugates: A Review of Approved Drugs and Their Clinical Level of Evidence

**DOI:** 10.3390/cancers15153886

**Published:** 2023-07-30

**Authors:** Pooja Gogia, Hamza Ashraf, Sidharth Bhasin, Yiqing Xu

**Affiliations:** 1Department of Hematology/Oncology, Maimonides Medical Center, Brooklyn, NY 11219, USA; yxu@maimonidesmed.org; 2Department of Internal Medicine, Overlook Medical Center, Summit, NJ 07901, USA; hamza.ashraf@atlantichealth.org; 3Department of Pulmonary Medicine, Saint Peter’s University Hospital, Rutgers-Robert Wood Johnson Medical School, New Brunswick, NJ 08901, USA; sbhasin@saintpetersuh.com

**Keywords:** ADC, antibody–drug conjugates, payloads, monoclonal antibodies, therapeutic ratio, histology-agnostic, bystander effect, Food and Drug Administration

## Abstract

**Simple Summary:**

Antibody–drug conjugates (ADC) have shown remarkable therapeutic activity in a wide range of hematologic and solid tumors. Herein, we discuss the mechanisms of action of ADCs, the desirable “bystander killing”, and the limitations associated with the design of each drug. We will also discuss the landmark clinical trials that demonstrated their treatment efficacy in each indication and that have been approved for clinical use by the Food and Drug Administration (FDA). We will also discuss unique side effects which are tied to the normal tissue expression of the antigen, the intrinsic toxicity of the payload, and the off-target toxicity. Finally, we will discuss some exciting new developments in the pipeline, including peptide–drug conjugates, immune-stimulating drug conjugates, and radioactive isotopes as payloads.

**Abstract:**

Antibody–drug conjugates (ADCs) are an innovative family of agents assembled through linking cytotoxic drugs (payloads) covalently to monoclonal antibodies (mAbs) to be delivered to tumor tissue that express their particular antigen, with the theoretical advantage of an augmented therapeutic ratio. As of June 2023, eleven ADCs have been approved by the Food and Drug Administration (FDA) and are on the market. These drugs have been added to the therapeutic armamentarium of acute myeloblastic and lymphoblastic leukemias, various types of lymphoma, breast, gastric or gastroesophageal junction, lung, urothelial, cervical, and ovarian cancers. They have proven to deliver more potent and effective anti-tumor activities than standard practice in a wide variety of indications. In addition to targeting antigen-expressing tumor cells, bystander effects have been engineered to extend cytotoxic killing to low-antigen-expressing or negative tumor cells in the heterogenous tumor milieu. Inevitably, myelosuppression is a common side effect with most of the ADCs due to the effects of the cytotoxic payload. Also, other unique side effects are specific to the tissue antigen that is targeted for, such as the cardiac toxicity with Her-2 targeting ADCs, and the hemorrhagic side effects with the tissue factor (TF) targeting Tisotumab vedotin. Further exciting developments are centered in the strategies to improve the tolerability and efficacy of the ADCs to improve the therapeutic window; as well as the development of novel payloads including (1) peptide–drug conjugates (PDCs), with the peptide replacing the monoclonal antibody, rendering greater tumor penetration; (2) immune-stimulating antibody conjugates (ISACs), which upon conjugation of the antigen, cause an influx of pro-inflammatory cytokines to activate dendritic cells and harness an anti-tumor T-cell response; and (3) the use of radioactive isotopes as a payload to enhance cytotoxic activity.

## 1. Introduction

Traditionally, the foundation of cancer treatment has been cytotoxic chemotherapy, surgery, and radiation therapy. However, in recent years, with the advent of immunotherapy and the ability to target specific molecular agents, the treatment of cancer has been revolutionized and is no longer limited only to the conventional chemotherapeutic agents [1]. In 1975, hybridoma technology helped to greatly advance the precision tumor targeting via monoclonal antibodies (mAbs). The concept of targeted delivery of drugs was first formulated in the early 1900s by Paul Ehrlich when he envisioned the concept of a magic bullet which would ensure that the drugs achieve their intended cellular targets while remaining inoffensive to the normal tissues [2]. Hereafter, an innovative conception of antibody–drug conjugate was proposed that conglomerates both the target specificity of mAbs and the anti-tumor properties of cytotoxic molecules, thereby improving the therapeutic window [3]. Nonetheless, it was only in the 1980s that the first ever clinical trials of ADCs were designed that unfortunately did not elicit satisfying results due to minimal efficacy and high toxicity [4]. Consequently, in the year 2000, a substantial success was achieved in the development of ADC with the advent of the anti-CD33-targeted agent gemtuzumab ozogamicin for adults with acute myeloid leukemia [5]. A decade later, the next ADC, brentuximab vedotin, was approved for the treatment of classical Hodgkin lymphoma and systemic anaplastic large cell lymphoma [6]. In 2013, the first approved ADC to treat solid tumors was the human epidermal growth factor receptor 2 (HER2)-targeted ADC, ado-trastuzumab emtansine (T- DM1), to treat metastatic breast cancer [7].

Currently, there are eleven FDA-approved antibody–drug conjugates available in the market. Furthermore, over 100 ADCs are in the diverse stages of development in clinical trials at the present time. Notably, a few novel ADCs have established substantial anti-tumor activity in many tumor types that share the expression of the targeted antigen, thus establishing a histology-agnostic activity of these compounds [8]. For instance, HER2 overexpressing breast, gastric and colorectal cancers, and Her-2 mutated lung cancers have demonstrated a significant response to the anti-Her2-targeting ADC, trastuzumab deruxtecan [9].

In this review, we will discuss mechanisms of action and key components in the development of ADCs. We will also discuss in detail the landmark trials that led to the approval of all the ADCs in sequence from their date of FDA approval, their unique toxicities, and the strength of the evidence per the National Comprehensive Cancer Network (NCCN) guidelines.

## 2. Antibody–Drug Conjugates

### 2.1. Mechanism of Action

Antibody–drug conjugates are comprised of an antibody, a cytotoxic payload, and a chemical linker [10]. Once the ADC reaches the target cells, the mAb component recognizes and binds to the cell surface antigens, and the ADC–antigen complex is then internalized within the cancer cell by endocytosis to form an early endosome, which, following a maturation, forms late endosomes and finally fuses with lysosomes. The cytotoxic drug payload is then released from the mAb via either a chemical reaction or enzyme digestion in the lysosomes, and exerts its cytotoxic effect, causing cell apoptosis or death [11].

In addition to the cytotoxic properties from the payload, the Fc portion of the monoclonal antibody aids in immune-related cytotoxicities, such as antibody-dependent cell mediated cytotoxicity (ADCC), antibody-dependent phagocytosis (ADP), and complement-dependent cytotoxicity (CDC) [12]. Genetic engineering technologies have advanced to enhance the effector function of the antibody in the Fc region [12]. Additionally, the binding of the antibody component of ADC with the specific antigen epitope of cancer cells can inhibit the downstream signal transduction of the antigen receptor [13]. For example, in T-DM1, the antibody trastuzumab binds to the HER2 receptor of cancer cells and inhibits the development of heterodimer between HER2 and HER1, HER3 or HER4, which blocks the signal transduction pathways of cell survival and proliferation to induce the cell apoptosis [14].

### 2.2. Key Components

Each component, as well as the conjugation methods of an ADC, can affect the final efficacy and safety of ADC. Key components of all the ADCs and adverse effects are listed in Table 1.

### 2.3. Target Antigen

An appropriate selection of target antigen is imperative to the design of an ADC. First, the antigen should be expressed, either exclusively or predominantly, in the tumor cells to reduce the off-target toxicity [15]. Secondly, the binding to the target antigen should ideally lead to the internalization of the antigen–antibody complex. Additionally, it should ideally be on the surface rather than intracellular for it to be recognized, and lastly, it should not be secretory, since a secreted antigen in the circulation would cause the undesirable ADC to bind outside of the tumor sites [16,17].

The target antigens of the FDA licensed ADCs are CD19, CD22, CD30, CD33, and CD79b in hematological malignancies and HER2, trop2, nectin4, tissue factor, and folate receptor alpha (FRα) in solid cancers as illustrated in Table 1.

### 2.4. Antibody

An ideal antibody moiety should also facilitate an effective internalization, have high antigen affinity, preserve long plasma half-life, and demonstrate low immunogenicity [18]. The mAb are large-sized and account for over 90% of the mass of any given ADC. This is favorable because it encounters reduced distribution or permeation into healthy tissue, including those normally functioning as metabolizing and eliminating organs [19]. However, no such problem is encountered at the tumor site as the vasculature in the tumor is characteristically leaky and allows the distribution and permeation of the ADCs to the tumor cells [20].

**Table 1 cancers-15-03886-t001:** ADCs with key components: targeted antigen (including expressed tissue), cytotoxic payloads, linker, drug antibody ratio, and adverse events (including unique toxicity and mechanism of toxicity).

ADC	Targeted Tumor Antigen and Expressed Tissue	Linker	Drug Antibody Ratio	Cytotoxic Payload	Adverse Events/Toxicities	Toxicity Mechanism
1. Gemtuzumab ozogamicin (Mylotarg)	CD33 (Expressed in myeloid stem cells, myeloblasts, monoblasts, granulocyte precursors, mast cells)	Cleavable acid-labile hydrazone (chemical)	2–3	Calicheamicin (cytotoxic antibiotic)	Infection Hemorrhage Increased transaminases Veno-occlusive disease Thrombocytopenia	Normal tissue expression of antigen: veno-occlusive disease, hemorrhage and hepatotoxity Payload-related: hepatic dysfunction Myelosuppression
2. Brentuximab vedotin (Adcetris)	CD30 (Expressed in a small subset of activated T and B lymphocytes, classical HL, anaplastic large cell lymphoma (ALCL), peripheral T-cell lymphoma (PTCL), adult T-cell leukemia/lymphoma; cutaneous T-cell lymphoma (CTCL); extra-nodal NK-T-cell lymphoma; and a variety of B-cell non-HLs, including diffuse large B-cell lymphoma, particularly EBV-positive diffuse large B-cell lymphoma)	Cleavable (enzymatic)	4	Monomethyl auristatin E (microtubule-targeting)	Peripheral sensory neuropathy Febrile neutropenia Thrombocytopenia Anemia Upper respiratory tract infection Mucositis	Payload-related: Peripheral sensory neuropathy Myelosuppression
3. Ado- Trastuzumab emtansine (Kadcyla)	Her-2 (Expressed in breast, colon, gastric, endometrial cancer cells)	Non-cleavable (thioether)	3–4	DM1, derivative of maytansine (microtubule-targeting)	Thrombocytopenia Neutropenia Hypertension Increased transaminases Pneumonitis Peripheral sensory neuropathy Ocular toxicity	Normal tissue expression of antigen: Cardiac toxicity (Decrease LVEF) Payload-related: Myelosuppression Increased transaminases Peripheral sensory neuropathy Off target toxicity: Interstitial pneumonitis
4. Inotuzumab ozogamicin (Besponsa)	CD22 (Expressed early during the ontogeny of B cells, and in the blasts of B cell acute lymphoblastic leukemia)	Cleavable acid-labile hydrazone linker (chemical)	5–7	Calicheamicin (cytotoxic antibiotic)	Thrombocytopenia Neutropenia Anemia Leukopenia Hemorrhage Pyrexia Headache Transaminases increased Gamma-glutamyltransferase increase Hyperbilirubinemia	Payload-related: Hepatic dysfunction Myelosuppression
5. Polatuzumab vedotin (Polivy)	CD79b (Expressed on over 90% of B-cell NHL malignancies)	Cleavable (Enzymatic)	3.5	Monomethyl auristatin E (microtubule-targeting)	Neutropenia Anemia Thrombocytopenia Peripheral neuropathy	Payload-related: Peripheral sensory neuropathy Myelosuppression
6. Enfortumab vedotin—(Padcev)	Nectin-4 (Over-expressed in several human cancers, including lung, gastric, ovarian and breast cancers)	Cleavable (Enzymatic)	3.8	Monomethyl auristatin E (microtubule-targeting)	Peripheral Neuropathy Decreased appetite Rash Alopecia Dysgeusia Dry Eye Dry Skin Pruritis	Normal tissue expression of antigen: Dysgeusia Payload-related: Peripheral sensory neuropathy
7. Fam-Trastuzumab deruxtecan (Enhertu)	Her-2 (Expressed in breast, colon, gastric, endometrial cancer cells)	Cleavable (Enzymatic)	7–8	Topoisomerase I inhibitor (exatecan derivative) (DNA-targeting)	Interstitial lung disease Neutropenia Anemia Thrombocytopenia Nausea Left Ventricular Dysfunction	Normal tissue expression of antigen: Cardiac toxicity (Decrease LVEF) Payload-related: Gastrointestinal toxicity Myelosuppression Off target toxicity: Interstitial pneumonitis
8. Sacituzumab govitecan (Trodelvy)	Trop-2 (Expressed in breast, cervix, colorectal, esophagus, gastric, certain lung cancers, squamous cell carcinoma of the oral cavity, ovary, pancreas, prostate, stomach, thyroid, urinary bladder, and uterus, also in several hematologic malignancies such as leukemia, extranodal nasal type lymphoma (ENK/TL) and NHL)	Cleavable acid-labile hydrazone (chemical)	7.6	SN-38 (active metabolite) of Irinotecan, topoisomerase-1 inhibitor (DNA-targeting)	Neutropenia Alopecia Anemia Vomiting Diarrhea Decreased Appetite Rash Hyperglycemia	Normal tissue expression of antigen: Skin rash Hyperglycemia Payload-related: Myelosuppression Diarrhea
9. Loncastuximab Tesirine (Zynlonta)	CD19 (Expressed in normal and neoplastic B cells, as well as follicular dendritic cells)	Cleavable (Enzymatic)	2.3	SG3199, alkylating agent (Pyrrolobenzodiazepine dimer) (DNA-targeting)	Thrombocytopenia Increased gamma-glutamyltransferase Neutropenia Anemia Hyperglycemia Transaminase elevation Hypoalbuminemia Fluid retention Edema Musculoskeletal pain	Payload-related: Increased gamma-glutamyltransferase Fluid retention Myelosuppression
10. Tisotumab vedotin (Tivdak)	Tissue factor (TF) (Expressed in cervical cancer, gastrointestinal, urogenital cancers, gliomas, melanomas, lung cancer, and breast cancer)	Cleavable (Enzymatic)	4	Monomethyl auristatin E (microtubule-targeting)	Anemia Leukopenia Peripheral Neuropathy Alopecia Epistaxis Conjunctival reactions Hemorrhage Increased creatinine Dry eye PT/INR/aPTT prolonged Rash	Normal tissue expression of antigen: Hemorrhagic complication and conjunctival reaction Payload-related: Peripheral sensory neuropathy Myelosuppression
11. Mirvetuximab soravtansine-gynx (Elahere)	Folate factor alpha (FRα) (Expressed in solid tumors such as ovarian, lung and breast cancers)	Cleavable Disufide bond based (chemical)	3.5	DM4 (maytansinoid derivative) (microtubule-targeting)	Reversible ocular (uveitis and keratopathy) Pneumonitis Peripheral neuropathy	Payload-related: Peripheral neuropathy Myelosuppression Off target toxicity Ocular toxicity

### 2.5. Linker

Diverse ADC properties are also impacted by linker chemistry, including specificity, stability, potency, and toxicity [21]. Depending on the methods of releasing the payload in cells, there are two types of linkers, including cleavable and non-cleavable [22]. The cleavable linkers are either chemically labile (hydrazone bond and disulfide bond) or enzymatically labile. Hydrazone linkers are generally stable in alkaline environments and are hydrolyzed in low pH environments, such as that in the lysosome and endosome. Hence, the cleavage of ADCs with hydrazone linkers occurs predominantly in the lysosome and endosome upon internalization, with occasional hydrolysis in the plasma, resulting in off-target, systemic toxicity [23]. Similarly, a disulfide bond linker can be stable in the plasma while specifically releasing the active payloads in the cancer cells with an elevated reductive glutathione level [24]. The enzyme sensitive linkers are sensitive to the lysosomal protease that is generally overexpressed in cancer cells, enabling an accurate drug release in the cells after internalization [25]. ADCs with non-cleavable linkers are resistant to chemical or enzymatic digestion in the plasma and will require complete degradation of the antibody within the late endosomes and lysosome to release the payload. Therefore, ADCs with non-cleavable linkers may have the lowest off-target systemic toxicity due to increased plasma stability [26,27], and thus they are most suitable in the treatment of tumors with homogenous antigen expression. Some of the ADCs have been engineered to have desirable “off-target effect” for “by-stander killing”, extending the cytotoxic effect to the low or negative antigen-expressing cells in the tumor proximity. For this mechanism to work, several characteristics of the ADC molecules are crucial: namely, a cleavable linker and a non-polar, freely membrane-permeable payload [28]. For instance, Trastuzumab deruxtecan (T-DXd) has a cleavable enzymatic linker as opposed to trastuzumab emtansine that has a thioether non-cleavable linker. Payloads such as deruxtecan, MMAE, or maytansinoid DM4 are cell membrane-permeable, and can diffuse out of the cells after cleavage in the lysosomes. Hence, by virtue of its component properties, TDXd has bystander effect, thus is effective even in low-Her2-harboring tumors, whereas trastuzumab emtansine due to its properties has a lower toxicity profile. Conversely, to reduce the undesirable systemic toxicity from payload molecules permeating out of the tumor cells, ionizable payloads (e.g., containing carboxylic acids) can be used [29]. As of now, trastuzumab emtansine is the only approved ADC that uses a non-cleavable linker, while the rest of ADCs are equipped with cleavable linkers. Modifications and new developments of linker bonds and structures are evolving constantly.

### 2.6. Cytotoxic Drugs

The cytotoxic payloads should ideally have the following properties:

High potency, in vitro high cytotoxic activity (sub nanomolar half maximal inhibitory concentration (IC50) value), high stability in the systemic circulation, sufficient solubility in the aqueous environment of antibody and biochemical properties to allow easier conjugation to the antibody, low immunogenicity, small molecular weight, and a long half-life [30,31].

In the current approved ADCs, there are mainly two classes of cytotoxic drugs used as payloads: the microtubule inhibitors or the DNA damaging agents.

Auristatins and maytansines payloads are both cytotoxic agents that work as tubulin inhibitors. Auristatin is a dolastatin synthetic analog. There are two auristatin derivatives: one is monomethyl auristatin E (MMAE) and the other is monomethyl auristatin F (MMAF). These two products differ structurally wherein the phenylalanine present at the C-terminus renders MMAF membrane-impermeable, whereas the MMAE can exit the cell and thus diffuse to nearby cells and kill them through the bystander effects [32]. The ADCs with MMAE payload are Brentuximab Vedotin, Polatuzumab Vedotin, Enfortumab Vedotin, and Tisotumab Vedotin, while Belantamab mafodotin uses MMAF as a cytotoxic payload (Belantamab is now withdrawn from the market).

Maytansinoids are natural cytotoxic agents isolated from the cortex of Maytenus serrata, which possesses a macrolide structure. Maytansionoid payloads, DM1 and DM4, are a component of Trastuzumab Emtansine and Mirvetuximab Soravtansin, respectively [32,33].

Calicheamicins, Pyrrolobenzodiazepines and topoisomerase inhibitors are DNA-damaging agents that act through DNA double strand breaks, crosslinking, and intercalation, respectively [34]. Both Gemtuzumab ozogamicin and Inotuzumab Ozogamicin have N-acetyl gamma calicheamicin as a payload. Calicheamicins belong to a class of potent anti-tumor antibiotics that cleave the DNA in a site-specific, double-stranded manner. Pyrrolobenzodiazepines are another class of antibiotics derived from Streptomyces species and is used as a cytotoxic payload in Loncastuximab Tesirine [34]. SN-38 and Deruxtecan are topoisomerase inhibitors that are the cytotoxic components of Sacituzumab Govitecan and Trastuzumab Deruxtecan, respectively [35].

The cytotoxic payloads used for are listed in Table 1.

### 2.7. Conjugation

In addition to the choice of the antibody, the linker, and the payload, the method of conjugation is also important for the successful structure of ADCs. The lysine and cysteine residues on the antibody provide the accessible reaction sites for conjugation [36,37]. A varying number (0–8) of small-molecule toxins may be attached to an antibody, as the conventional conjugation methods are random, resulting in a wide drug–antibody ratio (DAR) distribution [38]. The ideal DAR is 2–4. A low DAR can lower the efficacy, while a high DAR may increase the drug potency, but it also could negatively affect antibody structure, stability, and antigen binding, leading to faster clearance and decrease in overall clinical activity [39].

## 3. FDA Approved ADCs

### 3.1. Gemtuzumab Ozogamicin

Gemtuzumab ozogamicin (GO, also known as CMA-676) is a humanized anti-CD33 monoclonal antibody linked by an acid-labile hydrazone cleavable linker to the cytotoxic agent N-acetyl gamma calicheamicin that belongs to the family of enediyne antibiotics. The monoclonal antibody binds to the CD33 antigen, thereby leading to the internalization and release of the cytotoxic agent, which induces double-strand DNA breaks and cell death [40].

On 17 May 2000, GO received accelerated approval for the treatment of older patients with relapsed CD33-positive acute myeloid leukemia (AML), but the approval later was withdrawn based on safety and efficacy data in 2010 [41]. The original recommended dose was 9 mg/m^2^, a total of two doses administered 14 days apart. Fatal hepatotoxicity and veno-occlusive disease (VOD) in patients who received GO before or after hematopoietic stem cell transplantation (HSCT) was issued as a black box warning [42]. The withdrawal, however, was based on a randomized clinical trial evaluating GO 6 mg/m^2^ in combination with induction chemotherapy in patients ≤ 60 years with newly diagnosed AML. The trial failed to demonstrate a clinical benefit, but instead showed a higher rate of fatal induction toxicities compared with chemotherapy alone [43]. Recent pharmacokinetic studies suggest that the reason for the serious toxicity was that the original dose was too high [41].

Later, on 1 September 2017, the U.S. Food and Drug Administration approved GO (Mylotarg, Pfizer) for the treatment of relapsed or refractory (R/R) CD33-positive AML in adults as a monotherapy, as well as for newly diagnosed CD33-positive AML in adults and pediatric patients 1 month and older in combination with chemotherapy. The recommended dose is 3 mg/m^2^ [44].

For R/R CD33-positive AML, the single-arm phase II (MyloFrance 1) trial on 57 patients showed GO monotherapy resulted in a CR/CRp rate of 26% (95% confidence interval (CI) 16–40%) and the median relapse-free survival was 11.6 months [45].

For newly diagnosed de novo AML, a multicenter, randomized, open-label phase 3 study (ALFA-0701) of 280 patients aged from 50 to 70 years of age showed a significant event-free survival (EFS) benefit of 17.3 months for patients receiving GO with combination chemotherapy vs. 9.5 months for those receiving chemotherapy alone, with a hazard ratio of 0.56 (95% CI: 0.42, 0.76). There was no significant difference in the rate of complete remission (CR) (70.4% vs. 69.9%), but the median overall survival (OS) was 27.5 months [95% CI: 21.4–45.6] in the GO arm and 21.8 months (95%CI: 15.5–27.4) in the control arm [Hazard Ratio (HR), 0.81; 95%CI: 0.60–1.09; 2-sided *p* = 0.16) [46].

Furthermore, a multicenter, randomized study of 1022 patients (511 each arm) with newly diagnosed AML in patients of ages from 0 to 29 years (AAML0531 (NCT00372593)) showed that GO, in combination with chemotherapy, significantly improved 3-year EFS (53.1% vs. 46.9%; HR 0.83; 95% CI: 0.70 to 0.99; *p* = 0.04), but not 3-year OS (69.4% vs. 65.4%; HR 0.91; 95% CI: 0.74 to 1.13; *p* = 0.39). There was no difference in CR (88% vs. 85%; *p* = 0.15); however, a 3-year relapse risk (RR) was significantly less among GO (32.8% vs. 41.3%; HR, 0.73; 95% CI: 0.58 to 0.91; *p* = 0.006) [47]. Based on the EFS benefit, the FDA later extended the indication of GO for newly diagnosed CD33-positive acute myeloid leukemia (AML) to include pediatric patients 1 month and older, on 16 June 2020 [48].

### 3.2. Brentuximab Vedotin

Brentuximab vedotin (BV) is a CD30-directed ADC consisting of the chimeric IgG1 antibody cAC10, specific for human CD30 and a microtubule-disrupting agent, microtubule-binding auristatin, monomethyl auristatin E (MMAE), with a protease-cleavable linker [49]. The proteolytic cleavage after internalization releases MMAE, which induces cell cycle arrest and apoptosis by disrupting the microtubular network [50]. Due to the advantage of the bystander effect from the membrane permeability of MMAE, brentuximab vedotin is also effective in the histologically heterogenous tumor expression of CD30 [45].

On 19 August 2011, the U.S. Food and Drug Administration (FDA) provided accelerated approval of BV (Adcetris, Seattle Genetics) for two indications [51].

The first approved use of Brentuximab vedotin was in patients with Hodgkin lymphoma, who either relapsed after two or more prior lines of therapy, after autologous stem cell transplant (ASCT), or those who were not ASCT candidates. A phase II trial that involved 102 patients with relapsed and refractory Hodgkin lymphoma and treated with single-agent BV suggested an overall response rate (ORR) of 75% (95% CI, 64.9% to 82.6%), with CR in 34% of patients (95% CI, 25.2% to 44.4%). The median progression-free survival (PFS) was 5.6 months (95% CI, 5.0 to 9.0 months), the median duration of response in patients achieving a CR was 20.5 months, and the median OS was 22.4 months [52].

Second, it was approved for the treatment of patients with systemic anaplastic large-cell lymphoma (ALCL) after failure of at least one prior multi-agent chemotherapy regimen.

In a phase II study of 58 patients with relapsed or refractory systemic anaplastic large-cell lymphoma, who had been treated with single-agent BV, ORR was 86% (95% CI, 74.6% to 93.9%), with 57% CR. The median duration of overall response was 12.6 months, the median PFS was 13.3 months (95% CI, 6.9 months to NE), and the OS was not reached [53].

The approval was then expanded for classical Hodgkin lymphoma at a high risk of relapse after or progression after consolidation with ASCT [54]. AETHERA was a phase 3 clinical trial with single-agent BV administered every 3 weeks following ASCT, compared to a placebo for up to 16 cycles, that showed statistically significant improved PFS in patients receiving BV 42.9 months vs. 24.1 months, respectively (hazard ratio [HR] 0.57, 95% CI 0.40–0.81; *p* = 0.0013) [55].

The FDA granted regular approval of BV on 9 November 2017 for the treatment of CD30-expressing mycosis fungoides (MF) or primary cutaneous anaplastic large cell lymphoma (pcALCL), for patients who have received prior systemic therapy [56]. This was based on a phase 3, randomized, open-label, multicenter clinical trial called ALCANZA, in which 131 patients with MF or pcALCL who, after previously receiving one prior systemic therapy, then received BV. The objective response was achieved in 56.3% of patients (36/64 patients) with BV vs. 12.5% (8/64) with the physician’s choice seen after a median follow-up of 22.9 months (95% CI 29·1–58·4; *p* < 0.0001) [57].

Later, on 20 March 2018, BV in combination with chemotherapy was approved for use for the treatment of adult patients with newly diagnosed Stage III/IV classical Hodgkin lymphoma [58]. This approval was based on a trial called ECHELON-1, that included 1344 patients that were randomized 1:1 to receive either BV + AVD (doxorubicin, vinblastine, and dacarbazine) or ABVD (doxorubicin, bleomycin, vinblastine, and dacarbazine). After a median follow-up of 24.6 months, 2-year PFS in the BV + AVD vs. ABVD was 82.1% and 77.2%, respectively (HR 0.77; 95% CI, 0.60 to 0.98; *p* = 0.04), and there were 28 vs. 39 deaths with BV + AVD and ABVD, respectively (HR for interim overall survival is 0.73 [95% CI, 0.45 to 1.18]; *p* = 0.20) [59].

The FDA, on 16 November 2018, granted approval to BV with chemotherapy for newly diagnosed systemic anaplastic large cell lymphoma and other CD30-expressing peripheral T-cell lymphomas (PTCL) [60]. This was based on ECHELON-2 that randomized 452 patients to BV + CHP (cyclophosphamide, doxorubicin, and prednisone) and CHOP (cyclophosphamide, doxorubicin, vincristine, and prednisone) in a 1:1 ratio. The median PFS in the BV + CHP group was 48.2 months and in the CHOP group was 20.8 months (hazard ratio 0.71 [95% CI 0.54–0.93], *p* = 0.0110) [61].

On 10 November 2022, the FDA approved BV with doxorubicin, vincristine, etoposide, prednisone, and cyclophosphamide for previously untreated high-risk classical Hodgkin lymphoma (cHL) in pediatric patients more than 2 years of age. This was based on a trial of 600 patients randomized in a 1:1 ratio to brentuximab vedotin plus doxorubicin, vincristine, etoposide, prednisone, and cyclophosphamide [brentuximab vedotin + AVEPC] vs. A + bleomycin +V + E + P + C [ABVE-PC]. The 3-year EFS was 92.1% in the brentuximab vedotin group vs. 82.5% in the ABVE-PC group after a median follow-up of 42.1 months (HR 0.41; 95% CI, 0.25 to 0.67; *p* < 0.001) [62].

### 3.3. Ado-Trastuzumab Emtansine

Ado-trastuzumab emtansine is a HER2-targeted antibody–drug conjugate. The antibody component is the humanized anti-HER2 IgG1, and trastuzumab, and the small molecule cytotoxin is DM1. The linker is non-cleavable and hence stable in both the circulation and the tumor microenvironment; thus ado-trastuzumab emtansine, upon binding to the sub-domain IV of the HER2 receptor, undergoes lysosomal proteolytic degradation, resulting in the release of DM-1. The binding of DM-1 to tubulin then disrupts microtubule networks in the cell and hence leads to cell cycle arrest and apoptotic cell death [63].

The FDA, on 22 February 2013, approved Trastuzumab emtansine (Kadcyla, Roche) for patients with metastatic HER2-positive breast cancer, who had previously received trastuzumab and a taxane. [64]. This was based on the phase III EMILIA trial. In this trial, 991 patients with HER2-positive metastatic breast cancer were randomized to receive ado-trastuzumab emtansine vs. lapatinib with capecitabine. The study demonstrated an ORR of 43.6% vs. 30.8% (*p* < 0.001), a median PFS of 9.6 months vs. 6.4 months (HR 0.65; 95% confidence interval [CI], 0.55 to 0.77; *p* < 0.001), and the median overall survival of 30.9 months vs. 25.1 months; (HR 0.68; 95% CI, 0.55 to 0.85; *p* < 0.001) [65].

Later, on 3 May 2019, the FDA approved TDM1 as a single agent for adjuvant treatment of patients with HER2-positive breast cancers that have residual disease after receiving neoadjuvant trastuzumab-based therapy. This was based on a randomized open-label phase III trial KATHERINE [66]. This trial enrolled a total of 1486 patients that were randomized to either receive TDM1 or trastuzumab as an adjuvant treatment for HER2-positive early breast cancer that have residual disease. This study demonstrated a 3-year invasive disease-free survival (IDFS) benefit in patients who received ado-trastuzumab emtansine, compared with those who received trastuzumab (88.3% vs. 77%) (HR 0.50; 95% confidence interval, 0.39 to 0.64; *p* < 0.001) [67].

### 3.4. Inotuzumab Ozogamicin

Inotuzumab ozogamicin is comprised of an anti-CD22 humanized monoclonal antibody that is linked to calicheamicin, a cytotoxic antibiotic by an acid-labile hydrazone linker [68]. CD22 is an endocytic receptor that is a specific marker of B-cell acute lymphocytic leukemia (ALL) and is expressed in over 90% of patients with B-cell malignancies [69]. Once the antibody–drug conjugate binds to the CD22 receptor, the complex becomes internalized inside the targeted cell, triggering the release of calicheamicin, which induces cellular apoptosis by binding with the minor groove of double-helical DNA and causing site-specific double-stranded DNA cleavage [68,70].

Inotuzumab ozogamicin (Besponsa, Pfizer) was given FDA approval for the treatment of adults with relapsed or refractory B-cell precursor ALL on 17 August 2017 [71]. A phase 3 INTO-VATE ALL trial randomized a total of 326 patients to receive either inotuzumab ozogamicin or standard intensive chemotherapy. The CR rate was 80.7% (95% CI: 72.1, 87.7) vs. 29.4% (95% CI: 21.0, 38.8), the median PFS was 5.0 months [95% CI: 3.7, 5.6] vs. 1.8 months [95% CI: 1.5, 2.2] (hazard ratio, 0.45 [97.5% CI, 0.34 to 0.61]; *p* < 0.001), and the median overall survival was 7.7 months (95% CI: 6.0, 9.2) vs. 6.7 months (95% CI: 4.9 to 8.3) (hazard ratio, 0.45 [97.5% CI, 0.34 to 0.61]; *p* < 0.001) in inotuzumab ozogamicin vs. standard chemotherapy, respectively [68,72].

### 3.5. Polatuzumab Vedotin Piiq

Polatuzumab vedotin is a conjugate composed of an anti-CD79b monoclonal antibody linked via a protease-cleavable linker to monomethyl auristatin E (MMAE), which is a potent microtubule inhibitor, used similarly in the ADC of Brentuximab vedotin [72,73].

Polatuzumab vedotin, (Polivy, Roche) in combination with bendamustine and rituximab, was granted accelerated approval on 10 June 2019 for the treatment of relapsed or refractory diffuse large B-cell lymphoma after receiving two or more therapies [74]. This approval was based on an open-labeled phase 2 trial that included 80 patients with relapsed or refractory DLBCL after at least one prior regimen, who were randomized 1:1 to either P + BR vs. BR alone. This trial showed an ORR of 45% vs. 17.5%, a CR of 40% (95% CI: 25–57%) vs. 17.5% (*p* = 0.026), a median PFS of 9.5 *vs*. 3.7 months (HR, 0.36; 95% CI, 0.21 to 0.63; *p* < 0.001), and a median OS of 12.4 months vs. 4.7 months (HR, 0.42; 95% CI, 0.24 to 0.75; *p* < 0.002) with P + BR vs. BR alone after a median follow-up of 22.3 months [75].

### 3.6. Enfortumab Vedotin

Enfortumab vedotin (EV) is a nectin-4 targeted antibody–drug conjugate that is comprised of a human monoclonal antibody specific for nectin-4 and linked by a protease cleavable linker with monomethyl auristatin E (MMAE) [76]. Nectin-4 is a cell-adhesion molecule that is overexpressed in 97% of urothelial carcinomas and is associated with tumor growth and proliferation [77]. Directed delivery of the cytotoxic payload, monomethyl auristatin E, results in cell cycle arrest and apoptosis [78].

Enfortumab vedotin (Padcev, Seagen) was granted accelerated approval by the FDA on 18 December 2019 for use in adult patients with locally advanced or metastatic urothelial carcinoma who had previously received treatment with either a programmed death receptor-1 (PD-1) inhibitor or a programmed death ligand receptor-1 (PD-L1) inhibitor, along with a platinum-containing chemotherapy agent [79]. The agent was approved after a phase II trial consisting of 125 patients with locally advanced or metastatic urothelial cancer who had previously received treatment with either a PD-1 or PD-L1 inhibitor and a platinum-containing chemotherapy agent. The trial found that the objective response rate (ORR) was 44% (95% CI: 35.1, 53.2), with 12% having a complete response and 32% having a partial response, with a medium response duration of 7.6 months [80].

On 9 July 2021, the FDA granted regular approval for enfortumab vedotin to be used in adults with advanced or metastatic urothelial carcinoma who had been treated with either a PD-1 inhibitor or PD-L1 inhibitor and platinum-containing therapy, or those who were ineligible for cisplatin-containing therapy and had been treated with at least one prior line of treatment [81]. The regular approval was granted based on trial EV-301, which was done to corroborate the benefit of the aforementioned 2019 approval. EV-301 was a randomized phase III study and randomized 608 participants with advanced or metastatic urothelial cancer who received PD-1 or PD-L1 inhibitor and platinum-based therapy to single-agent Enfortumab vedotin or a single chemotherapy agent of the investigator’s choice (docetaxel, paclitaxel, or vinflunine). The trial revealed that enfortumab vedotin significantly lengthened survival and showed superior efficacy when compared to traditional single-agent chemotherapy, with the median overall survival and progressive-free survival in the enfortumab group being 12.9 months (95% CI: 10.6, 15.2) and 5.6 months (95% CI: 5.3, 5.8), compared with 9.0 months (95% CI:8.1, 10.7) and 3.7 months (95% CI:3.5, 3.9) in those receiving traditional chemotherapy (95% CI: 8.1, 10.7), respectfully. [65,70]. Additionally, the efficacy of enfortumab in adults with urothelial cancer who received PD-1 or PD-L1 inhibitor therapy, but were ineligible for platinum-containing therapy, was assessed in cohort 2 of EV-201 (n = 89), with the overall response rate determined to be 51% (95% CI: 39.8, 61.3) and with a median response duration of 13.8 months [76].

The FDA then granted accelerated approval to enfortumab vedotin plus pembrolizumab on 3 April 2023 for patients with locally advanced or metastatic urothelial carcinoma and who were not eligible for cisplatin-containing chemotherapy [82]. The approval was based on the cohort A/combined dose-escalation and cohort K of the phase 1b/2 EV-103/Keynote-869 (NCT03288545) trial. In the 121 treated patients, after a median follow-up of 44.7 months and 14.8 months in cohort A/combined dose-escalation and cohort K, respectively, the objective response rate was 68% (95% CI, 59–76%) with a complete response rate of 12%. The median DOR in the dose-escalation cohort plus cohort A was 22.1 months (range, 1.0+ to 46.3+) and was not yet reached (range, 1.2 to 24.1+) for cohort K [83,84].

There is currently an ongoing phase 3 trial EV-302 trial (NCT04223856) to confirm this approval.

### 3.7. Trastuzumab Deruxtecan

Fam-trastuzumab deruxtecan (T-DXd) is an antibody–drug conjugate composed of an anti-HER2 antibody trastuzumab, a cleavable tetrapeptide-based linker, and a cytotoxic topoisomerase I inhibitor (an exatecan derivative) [85]. Extensive preclinical studies have predicted T-DXd to be effective in tumor cells with both a high or low Her2 expression status or Her2 mutated tumor cells. There are multiple explanations for the same. Firstly, the ADC has a high drug–to–antibody ratio of 8, and therefore it carries a high potency payload; secondly, the released payload has a high membrane permeability, allowing it to enter neighboring tumor cells, resulting in a bystander effect; and thirdly, it has a high stability in plasma due to the novel tetrapeptide-based linker [86]. In addition, the payload has a short half-life, thereby minimizing the systemic toxicity [80]. Moreover, in vitro and in vivo pharmacologic activities have shown that T-DXd showed potent anti-tumor activity against tumor cells that were resistant to T-DM1, which is likely due to a higher susceptibility of T-DM1 to p-glycoprotein mediated efflux, while T-DXd was not subject to this drug efflux mechanism. Hence, Trastuzumab deruxtecan has shown significant anti-tumor activity in several HER2-expressing cancers, including HER2-positive and HER2-low breast cancer, HER2-positive gastric cancer, and NSCLC with HER-2-activating mutations, regardless of Her2 protein expression and HER2-overexpressing colorectal cancer [87].

Fam-trastuzumab deruxtecan (Enhertu, Daiichi Sankyo) received accelerated FDA approval in December 2019 for patients with unresectable or metastatic HER2-positive breast cancer, who had previously been treated with two or more prior HER2 targeted regimens in the metastatic setting [88]. This was based on the results of DESTINY-Breast01 trial that enrolled 253 patients with HER2-positive, unresectable, or metastatic breast cancer, and who had received previous treatments including trastuzumab emtansine. In 184 patients who received trastuzumab deruxtecan after a median of six previous treatments, the ORR was 60.9% (95% [CI], 53.4 to 68.0), the median PFS was 16.4 months (95% CI, 12.7 to not reached), and the overall survival was 86.2% (95% CI, 79.8 to 90.7) at 12 months; the median overall survival was not reached [89].

On 4 May 2022, the FDA granted regular approval to trastuzumab deruxtecan for patients with unresectable or metastatic HER2-positive breast cancer who had received a prior HER2- targeted regimen in any setting [90]. This was based on DESTINY-Breast03, a phase 3, multicenter, randomized trial that enrolled 524 patients to compare trastuzumab deruxtecan vs. trastuzumab emtansine in patients with HER2-positive metastatic breast cancer, previously treated with trastuzumab and a taxane. The ORR was 79.7% (95% CI, 74.3 to 84.4) vs. 34.2% (95% CI, 28.5 to 40.3) in trastuzumab deruxtecan vs. trastuzumab emtansine. At 12 months, the percentage of those who were alive without disease progression was 75.8% in the trastuzumab deruxtecan group (95% confidence interval [CI], 69.8 to 80.7) vs. 34.1% with trastuzumab emtansine (95% CI, 27.7 to 40.5) (HR for progression or death from any cause, 0.28; 95% CI, 0.22 to 0.37; *p* < 0.001), and the percentage of patients who were alive at 12 months was 94.1% (95% CI, 90.3 to 96.4) vs. 85.9% (95% CI, 80.9 to 89.7), respectively (HR for death, 0.55; 95% CI, 0.36 to 0.86; *p* value not reached) [91].

Trastuzumab deruxtecan was approved for the treatment of locally advanced or metastatic HER2-positive gastric or gastroesophageal junction adenocarcinoma in patients who had received a prior trastuzumab-based regimen on 15 January 2021 [92]. This was based on DESTINY-Gastric01, a multicenter, randomized trial of patients with HER2-positive locally advanced, metastatic gastric, or GEJ adenocarcinoma, and who progressed on at least two prior regimens. A total of 125 patients were randomized to receive T-DXd, while 62 received chemotherapy of the physician’s choice. This study demonstrated that the ORR was 51% vs. 14% (*p* < 0.001), and the median OS was 12.5 and 8.4 months (HR for death, 0.59; 95% confidence interval, 0.39 to 0.88; *p* = 0.01) in the T-DXd and chemotherapy groups, respectively [93].

On 5 August 2022, the FDA further approved T-DXd for unresectable or metastatic HER2-low as demonstrated by IHC 1+ or IHC 2 + /ISH− breast cancer that had received prior chemotherapy [94]. The DESTINY-Breast04 trial randomized 557 patients with unresectable or metastatic HER2-low breast cancer in a 2:1 ratio to receive T-DXd or the physician’s choice of chemotherapy. This study demonstrated an objective response rate of 52.3% (95% CI, 47.1 to 57.4) vs. 16.3% (95% CI, 11.3 to 22.5); the median PFS was 9.9 months and 5.1 months (HR 0.50; *p* < 0.001), and the OS was 23.4 months and 16.8 months, (HR 0.64; *p* = 0.001) in the T-DXd and chemotherapy arms, respectively [95].

The FDA extended and granted accelerated approval for unresectable or metastatic non-small cell lung cancer (NSCLC) with activating HER2 (ERBB2) mutations after progressing on prior therapy on 11 August 2022 [96]. This accelerated approval based on the DESTINY-Lung02 showed an ORR of 58% (95% CI: 43, 71) and a median DOR of 8.7 months (95% CI: 7.1, not estimable [NE]) in patients with HER2-mutant disease [87].

### 3.8. Sacituzumab Govitecan

Sacituzumab govitecan is an ADC that includes an antibody targeted against Trop-2 coupled with a toxin known as SN-38 by an acid-labile hydrazone cleavable linker [97]. Trop-2 is a transmembrane glycoprotein that is overexpressed in an extensive array of solid tumors and is a formidable target for cancer treatment due to its augmented expression in tumor cells as compared to nontumor cells. SN-38, which is the active metabolite in the chemotherapeutic agent Irinotecan, is a topoisomerase-1 inhibitor [98]. Once administered, the ADC binds to the Trop-2 on the targeted tumor cell and prompts the release of SN-38 to elicit DNA damage, subsequently leading to cell cycle arrest. Due to the membrane-permeable nature of SN-38, it can stimulate anti-tumor effects in nearby cells as well without being internalized, to exert a bystander effect [97].

On 22 April 2020, sacituzumab govitecan (Trodelvy, Immunomedics) was granted accelerated approval for the treatment of metastatic triple-negative breast cancer in adult patients who previously had received two or more treatments for metastatic disease [99]. The approval was granted based on IMMU-132-01, a phase I/II clinical trial consisting of 108 participants with a diagnosis of metastatic triple-negative breast cancer and had previously received at least two prior systemic therapies. The ORR and DOR were found to be 33.3% (95% CI: 24.6, 43.1) and 7.7 months (95% CI, 4.9, 10.8), respectively. Additionally, the clinical benefit rate was 45.4%, the median PFS was 5.5 months (95% CI: 4.1, 6.3), and OS was 13 months (95% CI, 11.2, 13.7) [100].

On 7 April 2021, sacituzumab govitecan was granted regular approval by the FDA for patients with unresectable locally advanced or metastatic triple-negative breast cancer who had previously received at least two prior systemic therapies, with at least one treatment being for metastatic disease [101]. The approval was granted based on a randomized trial including 468 patients with unresectable metastatic triple-negative breast cancer who had relapsed post-treatment with at least two prior chemotherapy agents, and who were randomized to receive Sacituzumab or a single-agent therapy of a physician’s choice. The percentage of objective response was 35% with sacituzumab govitecan and 5% with chemotherapy, and the median PFS was 5.6 months (95% confidence interval [CI], 4.3 to 6.3) in patients receiving sactiuzumab govitecan, compared with 1.7 months (95% CI: 1.5 to 2.6) in those receiving a single-agent chemotherapy agent. The median OS was noted as 12.1 months (95% CI: 10.7 to 14) in patients receiving Sacituzumab govitecan compared to 6.7 months (95% CI: 5.8 to 7.7) in those receiving chemotherapy (HR for death, 0.48; 95% CI, 0.38 to 0.59; *p* < 0.001) [102].

On 13 April 2021, the FDA granted accelerated approval for sacituzumab govitecan for patients with advanced or metastatic urothelial cancer who previously underwent treatment with either a PD-1 or a PD-L1 inhibitor [103]. The safety and efficacy of the agent were assessed in the phase II TROPHY trial, which included 113 participants with locally advanced or metastatic urothelial cancer, who had previously received treatment with either a PD-1 or PD-L1 inhibitor. The objective response rate was found to be 27.7% (95% CI:19.9, 36.9), and the medium duration of response was noted to be 7.2 months (95% CI: 4.7, 8.6), with a median PFS of 5.4 months (95% CI, 3.5 to 7.2 months) and an OS of 10.9 months (95% CI, 9.0 to 13.8 months) [104].

On 3 February 2023, the FDA extended approval to sacituzumab govitecan for patients with hormone-positive and Her-2/neu-negative metastatic breast cancer [105]. This approval was based on the TROPiCS-02 trial, a multicenter, open-label, phase III study that randomly assigned 543 patients to sacituzumab govitecan vs. physicians’ choice of chemotherapy in patients who had received between two and four prior lines of therapy. There was a median progression-free survival of 5.5 months in the sacituzumab govitecan arm and 4.0 months in the chemotherapy arm (HR = 0.66, *p* = 0.0003), and although the overall survival data were immature, there was a trend towards improvement with sacituzumab govitecan with no new safety concerns [106].

### 3.9. Loncastuximab Tesirine-Lpyl

Loncastuximab tesirine is an ADC that is composed of an antibody against CD19 which is linked through a cleavable enzymatic type linker with SG3199, a cytotoxic alkylating agent [107]. SG3199 is a synthetic pyrrolobenzodiazepine dimer that has a potent cytotoxic effect by promoting the formation of DNA interstrand cross-links and subsequently halting cell division [108].

Loncastuximab tesirine (Zynlonta, ADC Therapeutics) received accelerated approval in 23 April 2021 for patients with diffuse large B-cell lymphoma (DLBCL), after progressing on two or more lines of systemic therapy [109]. A phase II clinical trial LOTIS-2 that randomized 145 patients with relapsed or refractory DLBCL suggested an ORR of 48.3% (95% CI: 39.9, 56.7), a CR rate of 24.1% (95% CI: 17.4, 31.9) and a median DOR of 10.3 months after a median follow-up of 7.3 months [110].

### 3.10. Tisotumab Vedotin

Tisotumab vedotin is an ADC that is made up of a monoclonal antibody targeted against tissue factor (TF-011), a cleavable mc-VC-PABC linker and MMAE, the same cytotoxic agent used in Brentuximab vedotin, Polatuzumab vedotin, and enfortumab vedotin [111]. A study published in 2018 revealed that tissue factor is heavily expressed in cervical cancer tissues compared to adjacent samples of normal cervical tissues. Once tisotumab vedotin binds to the tissue factor, cytotoxic MMAE is delivered into the cell, which blocks tubulin polymerization and halts cell division. It exhibits a bystander cytotoxic effect due to its constituent properties of its payload’s membrane permeability, MMAE, and a cleavable linker. Preclinical studies have also demonstrated ADCC activity of the antibody [112,113].

On 20 September 2021, the FDA granted approval of tisotumab vedotin (Tivdak, Genmab/Seagen) for adult patients with either recurrent or metastatic cervical cancer with disease progression, despite receiving chemotherapy [112]. The approval was based on innovaTV 204, a phase 2 clinical study that included 101 adult patients who had recurrent or metastatic squamous cell, adenocarcinoma, or adenosquamous cervical cancer, and had received two or less systemic therapies for metastatic disease. The ORR was 24% (95% CI: 15.9, 33.3), with a 6.9% CR. The median DOR was 8.3 months (95% CI: 4.2, not reached) [113].

### 3.11. Mirvetuximab Soravtansine-Gynx

Mirvetuximab soravtansine comprises of a folate receptor α (FRα)-binding antibody and a tubulin-targeting cytotoxic agent maytansinoid DM4 linked by a cleavable disulfide linker. Upon binding to the antigen, the ADC is internalized, and DM4 is released. The antimitotic agent DM4 subsequently suppresses the microtubule, resulting in cell cycle arrest and apoptosis. The released free DM4, and its metabolites, can diffuse out from the antigen-positive tumor cells into neighboring cells and kill them in an antigen-independent manner, thereby taking advantage of the ‘bystander’ killing [114].

On 14 November 2022, the FDA granted mirvetuximab soravtansine (Elahere, ImmunoGen) accelerated approval for adults with folate receptor α (FRα)–positive, platinum-resistant epithelial ovarian, fallopian tube, or primary peritoneal cancer, and for which they had received between 1 and 3 previous systemic treatment regimens [115]. The accelerated approval is based on the phase II single-arm study SORAYA that evaluated 105 patients. The ORR was 32.4% (95% CI, 23.6–42.2%), including a 4.8% CR rate, the median DOR was 6.9 months (95% CI, 5.6–9.7), PFS was 4.3 months (95% CI, 3.7–5.2), and the median OS was 13.8 months (95% CI, 12.0-not estimable) [116].

## 4. ADCs Which Obtained FDA Approval and then Were Withdrawn from the Market

### 4.1. Belantamab Mafodotin-Blmf

Belantamab mafodotin is an ADC comprised of an IgG antibody targeted against B-cell maturation antigen (BCMA) and conjugated with monomethyl auristatin F (MMAF) through a non-cleavable maleimidocaproyl (mc) linker [117]. BCMA is a cell surface protein that is heavily expressed on plasma cells that are malignant in nature, and therefore, it is a promising target for the treatment of multiple myeloma [118]. Once belantamab mafodotin-blmf binds BCMA, the complex is internalized and degraded in lysosomes, triggering the release of the cytotoxic MMAF and leading to cell cycle arrest and apoptosis [117].

Belantamab mafodotin (Blenrep, GSK) was granted accelerated approval on 5 August 2020 by the FDA for use in adult patients with either relapsed or refractory multiple myeloma who have received at least four prior treatments which must include an anti-CD38 monoclonal antibody, an immunomodulary agent, and a proteasome inhibitor [119]. The approval was granted based on phase II DREAMM-2, a clinical trial which demonstrated an overall response rate of 31% in patients with relapsed or refractory multiple myeloma who had undergone three or more lines of therapy [120].

However, in November of 2022, the drug manufacturer GSK began the process for the removal of belantamab from the US market due to a request made by the FDA [121]. The FDA request was based on the DREAMM-3 confirmatory trial, which assessed belantamab vs. pomalidomide plus low-dose dexamethasone in patients with relapsed or refractory multiple myeloma. The median progression-free survival (PFS) was 11.2 months in the belantamab arm vs. 7 months in the pomalidomide arm (95% CI, 0.72–1.47), which did not meet the primary end point of PFS [122]. Additional clinical trials involving belantamab including the DREAMM-7 and DREAMM-8 phase III, which are still ongoing with results expected in the first half of 2023 [121].

### 4.2. Moxetumomab Pasudotox-Tdfk

Moxetumomab pasudotox is actually an immunotoxin categorized as an ADC that consists of an anti-CD22 monoclonal antibody covalently linked to a 38 kDa fragment of the Pseudomonas exotoxin A (PE38). The Fv portion of the immunotoxin binds to CD22, and once internalized, induces apoptotic death by catalyzing the ADP ribosylation of the diphthamide residue in elongation factor-2 (EF-2).

Unlike other ADCs, Moxetumomab is a recombinant immunotoxin that is not designed by chemical conjugation of the antibody to toxin [123].

Moxetumomab pasudotox (Lumoxiti, AstraZeneca) was initially granted FDA approval in September of 2018 for use in patients with relapsed or refractory hairy cell leukemia who had received at least two prior systemic therapies, including treatment with a purine nucleoside analog [124]. However, in November of 2022 it was announced that moxetuomomab would be withdrawn from the US market in July of 2023. The announcement, which was made by AstraZeneca, emphasized that the removal from the US market was not due to safety or efficacy concerns, but as a result of insufficient use. There had been a low clinical uptake of the drug since it was granted approval due to the availability of other treatment options and due to the intricacy of administration and safety monitoring requirements for patients [125,126]. In addition, a post-marketing PROXY study of moxetumomab was terminated [126].

Table 2 summarizes the indications, approval dates, dosage, administration, NCCN recommendation category, and landmark trials leading to the approvals of ADCs.

**Table 2 cancers-15-03886-t002:** ADCs with their indications, approval dates, dosage, administration, NCCN recommendation category, and landmark trials leading to their approvals.

ADC	FDA Approved Indications	FDA Approval Dates	Dose	NCCN Guideline Category	Clinical Trial
1. Gemtuzumab ozogamicin (Mylotarg)	Newly diagnosed CD33-positive acute myeloid leukemia (AML) to include pediatric patients 1 month and older	1 September 2017 16 June 2020 (Approval for pediatric patients: 1 month and older)	*Induction:* 3 mg/m^2^ (up to one 4.5 mg vial) on days 1, 4, and 7 in combination with daunorubicin and cytarabine. *Consolidation:* 3 mg/m^2^ on day 1 (up to one 4.5 mg vial) in combination with daunorubicin and cytarabine.	Category 2B [29]	Phase III study (ALFA-0701) Phase III (AAML0531)-for pediatric patients.
	Relapsed or refractory (R/R) CD33-positive AML	1 September 2017	3 mg/m^2^ on days 1, 4, and 7.	Category 2B [29]	Phase II (MyloFrance 1)
2. Brentuximab vedotin (Adcetris)	Classical Hodgkin lymphoma (cHL) after failure of ASCT or after failure of at least two prior multi-agent chemotherapy regimens in patients who are not ASCT candidates	19 August 2011 (Accelerated approval)	1.8 mg/kg by intravenous infusion every 3 weeks. (Maximum 16 cycles)	Category 2A [43]	Phase II
	cHL at high risk of relapse or progression as post-autologous hematopoietic stem cell transplantation (auto-HSCT) consolidation	17 August 2015	1.8 mg/kg by intravenous infusion every 3 weeks, starting 30–45 days after transplantation. (Maximum 16 cycles) (Initiate within 4–6 weeks post-auto-HSCT or upon recovery from auto-HSCT)	Category 2A [43]	Phase III (AETHERA)
	cHL previously untreated stage III or IV in combination with chemotherapy	20 March 2018	1.2 mg/kg by intravenous infusion every 2 weeks. (Maximum for 12 doses)	Category 2A [43]	Phase III (ECHELON-1)
	Relapsed or refractory systemic anaplastic large-cell lymphoma	19 August 2011 (Accelerated approval)	1.8 mg/kg by intravenous infusion every 3 weeks. (Maximum 16 cycles)	Category 2A [44]	Phase II
	MF or pcALCL who had previously received one prior systemic therapy	9 November 2017	1.8 mg/kg by intravenous infusion every 3 weeks until intolerance or disease progression.	Category 2A [44]	Phase III (ALCANZA)
	Previously untreated systemic anaplastic large cell lymphoma (sALCL)	16 November 2018	1.8 mg/kg by intravenous infusion every 3 weeks.	Category 1 [44]	Phase III (ECHELON-2)
	Previously untreated other CD30-expressing peripheral T-cell lymphomas (PTCL), including angioimmunoblastic T-cell lymphoma, and PTCL not otherwise specified	16 November 2018	1.8 mg/kg by intravenous infusion every 3 weeks with each cycle of chemotherapy for a maximum of 6 to 8 cycles.	Category 2A [44]	Phase III (ECHELON-2)
	Pediatric patients with previously untreated high-risk classical Hodgkin lymphoma	10 November 2022	1.8 mg/kg by intravenous infusion every 3 weeks with each cycle of chemotherapy for a maximum of 5 doses.	Not updated	Phase III (Study 7, AHOD1331, NCT02166463)
3. Ado- Trastuzumab emtansine (Kadcyla)	HER2-positive, metastatic breast cancer who have received prior therapy	22 February 2013	3.6 mg/kg intravenous over 30 to 90 min on day 1 of a 21-day cycle until progression of disease or unacceptable toxicity	Category 2A [50]	Phase III (EMILIA trial)
	HER2-positive early breast cancer as a single agent, for the adjuvant treatment of patients with residual invasive disease after neoadjuvant taxane and trastuzumab-based treatment	3 May 2019	3.6 mg/kg intravenously day 1 of a 21-day cycle for 14 cycles.	Category 1 [50]	Phase III (KATHERINE trial)
4. Inotuzumab ozogamicin (Besponsa)	Adults with relapsed or refractory B-cell precursor acute lymphoblastic leukemia (ALL)	17 August 2017	Initial: 1.8 mg/m^2^ in 3 divided doses on day 1 (0.8 mg/m^2^), day 8 (0.5 mg/m^2^), and day 15 (0.5 mg/m^2^). If CR/Cri:1.5 mg/m^2^ in 3 divided doses of 0.5 mg/m^2^ on days 1, 8, and 15. No CR or Cri:1.8 mg/m^2^ in 3 divided doses with 0.8 mg/m^2^ on day 1, and 0.5 mg/m^2^ on days 8 and 15.	Category 2A [55,56]	Phase III (INTO-VATE ALL)
5. Polatuzumab vedotin piiq (Polivy)	Adult patients in combination with bendamustine and rituximab for the treatment of adult patients with relapsed or refractory diffuse large B-cell lymphoma, not otherwise specified, after at least two prior therapies.	10 June 2019 (accelerated approval)	1.8 mg/kg every 21 days for 6 cycles in combination with bendamustine and a rituximab.	Category 2B [64]	Phase II trial
6. Enfortumab vedotin -ejfv (Padcev)	Advanced or metastatic urothelial cancer in patients who have previously received a PD-1 or PD-L1 inhibitor and a platinum-containing chemotherapy	9 July 2021	1.25 mg/kg (up to a maximum of 125 mg for patients ≥ 100 kg) administered as an intravenous infusion over 30 min on days 1, 8, and 15 of a 28-day cycle until disease progression or unacceptable toxicity.	Category 2A [71]	Phase III study (EV-301)
	Advanced or metastatic urothelial cancer in patients who are ineligible for cisplatin-containing therapy and have previously received one or more prior lines of therapy	9 July 2021		Category 2A [71]	Phase III study (EV-301)
	Locally advanced or metastatic urothelial carcinoma who are not eligible for cisplatin-containing chemotherapy as a first line in combination with pembrolizumab	3 April 2023 (Accelerated approval)	1.25 mg/kg intravenously over the course of 30 min on days 1 and 8 of each 21-day treatment cycle followed by pembrolizumab, intravenously at a dose of 200 mg on day 1 of each cycle about 30 min after enfortumab vedotin.	Not yet updated	Phase Ib/II (EV-103/Keynote-869)
7. Fam-Trastuzumab deruxtecan-nxki (Enhertu)	Unresectable or metastatic HER2-positive breast cancer who have received a prior anti-HER2-based regimen either in the metastatic setting, or in the neoadjuvant or adjuvant setting and have developed disease recurrence during or within 6 months of completing therapy.	4 May 2022	5.4 mg/kg IV day 1 every 21 days till disease progression or unacceptable toxicity.	Category 1 [80]	Phase III (DESTINY- Breast03)
	Unresectable or metastatic HER2-low (IHC 1+ or IHC 2 + /ISH-) breast cancer who have received a prior chemotherapy in the metastatic setting or developed disease recurrence during or within 6 months of completing adjuvant chemotherapy	5 August 2022		Category 1 [80]	Phase III (DESTINY- Breast04)
	Locally advanced or metastatic HER2-positive gastric or gastroesophageal junction adenocarcinoma who have received a prior trastuzumab-based regimen	15 January 2021	6.4 mg/kg IV day 1 every 21 days till disease progression or unacceptable toxicity.	Category 2A [81]	Phase II (DESTINY-Gastric01)
	Unresectable or metastatic non-small cell lung cancer (NSCLC) whose tumors have activating HER2 (ERBB2) mutations and who have received a prior systemic therapy	11 August 2022 (accelerated approval)	5.4 mg/kg IV day 1 every 21 days till disease progression or unacceptable toxicity.	Category 2A [82]	Phase II (DESTINY-Lung02)
8. Sacituzumab govitecan -hziy (Trodelvy)	Adult patients with unresectable locally advanced or metastatic triple-negative breast cancer who have previously received two or more systemic therapies, with at least one therapy for metastatic disease	7 April 2021	10 mg/kg once weekly on days 1 and 8 of a 21-day treatment cycle until disease progression or unacceptable toxicity.	Category 2A [90,91]	Phase III (ASCENT)
	Adult patients with locally advanced or metastatic urothelial cancer who have previously received a PD-1 or PD-L1 inhibitor	13 April 2021		Category 2A [90,92]	Phase II (TROPHY trial)
	Adult patients with unresectable, locally advanced or metastatic, hormone receptor (HR)-positive, HER2-negative breast cancer who have received endocrine-based therapy and at least two additional systemic therapies in the metastatic setting	3 February 2023		Not yet updated	Phase III (TROPiCS-02)
9. Loncastuximab Tesirine-lpyl (Zynlonta)	Adult patients with relapsed or refractory large B-cell lymphoma after at least two lines of systemic therapy, including diffuse large B-cell lymphoma not otherwise specified, diffuse large B-cell lymphoma arising from low grade lymphoma, and high-grade B-cell lymphoma	23 April 2021 (Accelerated approval)	0.15 mg/kg every 3 weeks for 2 cycles, administered on day 1 of each cycle (every 3 weeks), and 0.075 mg/kg every 3 weeks for subsequent cycles.	Category 2A [104,106,107]	Phase II (LOTIS-2)
10. Tisotumab vedotin-tftv (Tivdak)	Adult patients with recurrent or metastatic cervical cancer with disease progression on or after chemotherapy	20 September 2021(Accelerated approval)	2 mg/kg given as an intravenous infusion over 30 min every 3 weeks until disease progression or toxicity.	Category 2A [113,114]	Phase II (InnovaTV 204)
11. Mirvetuximab soravtansine-gynx (Elahere)	Folate receptor α positive, platinum-resistant epithelial ovarian, fallopian tube, or primary peritoneal cancer for patients who have received 1 to 3 previous systemic treatment regimens	14 November 2022 (Accelerated approval)	6 mg/kg intravenous administered once every 3 weeks until disease progression or unacceptable toxicity.	Not updated	Phase III single- arm study (SORAYA)

## 5. Unique Toxicities of ADCs

Although the conceptualization of ADCs is ideal, for achieving a target therapeutic index, the appropriate combination of a specific mAb, linker, and toxic payload has been quite challenging. Each of the component of ADC contributes to its toxicity profile. Specifically, we identified three categories of mechanisms for the unique toxicities of ADCs.

The first mechanism of the unique toxicity is its targeting of the normal tissue expression of the antigen, which is also called the “on-target toxicity”. For example, Tisotumab vedotin has a unique side effect of hemorrhagic complications, which is due to its binding to tissue factor (TF), which is a primary initiator of blood coagulation after vascular injury [111]. In addition, TF is expressed in the conjunctiva, and therefore, conjunctival inflammation is a consequence of direct delivery of MMAE to the tissue factor-expressing cells [127]. Other examples are dysgeusia caused by enfortumab vedotin, which is due to the expression of nectin-4 in the salivary glands [128], and skin rash and hyperglycemia from Sacituzumab govitecan due to Trop-2 expression [129] in the skin and pancreas, respectively.

Fortunately, the off-target antigen expression and antigen–antibody interaction in the above examples are not very strong and can be managed with precautions such as using anti-inflammatory eye drops and cold compression during infusion of Tisotumab vedotin [127].

However, some off-target expression can be significant for example, CD33 expression can be abundant in the peripheral mature myeloid cells. In an initial study of Gemtuzumab ozogamicin, the ADC that targets CD33, used a high dose of 9 mg/kg, which was shown to saturate all CD33-expressing myeloid cells and leukemia cells [130]. That strategy was associated with severe toxicity of myelosuppression, venous occlusive disease (VOD), and early death. Further studies exploring a smaller dose, repeated administration, and combinations with other chemotherapy agents have proven to be the optimal use of this ADC [131,132,133].

The second mechanism is that the different properties of the cytotoxic payload dictate their specific toxicities, with myelosuppression being the most common. Specifically, calicheamicin is associated with thrombocytopenia and hepatic dysfunction; MMAE, a microtubule-targeting drug, causes peripheral neuropathy; DM1, another microtubule-targeting agent, induces gastrointestinal effects, thrombocytopenia, and neutropenia and SN38 and deruxtecan, topoisomerase 1 inhibitors, are associated with gastrointestinal side effects [17]. SG3199, an alkylating agent (Pyrrolobenzodiazepine dimer) is associated with elevation of γ-glutamyl transpeptidase and fluid retention, including pleural effusion likely due to direct vascular injury [134].

Thirdly, besides the direct cytotoxicity by the payload, its release in the bloodstream and retention in certain organs is also implicated as the reason for organ-specific “off-target” toxicities. Interstitial Pneumonitis (ILD) caused by Trastuzumab emtansine and trastuzumab-deruxtecan are theorized to be due to these off-target effect of ample blood supply and longest retention time in the lungs [135]. Another unique side effect is ocular toxicity seen with multiple ADCs. This is common with ADCs carrying maytansinoids, i.e., DM1 and DM4, which due to its off-target effect, accumulates in the ocular space and causes side effects [136].

## 6. Challenges

The availability of a tumor-specific antigen and the development of a monoclonal antibody with high specificity and affinity is challenging [19]. Various tumor-specific antigens have been discovered, comprising glycoproteins, an extracellular matrix, and cell surface proteins [19]. However, despite the discovery of these tumor-specific antigens, the following difficulties persist. Firstly, high antigen affinity does not necessarily have high tumor penetration. Secondly, the distribution of cell surface target antigen expression defines the therapeutic window and a high antigen expression level in a tumor does not necessarily guarantee that an ADC will be highly effective [137,138].

ADCs have complex pharmacokinetic and pharmacodynamic profiles, which poses a challenge in designing them. Different factors affect the clearance of each of the component of an ADC. Antibody clearance is mediated by the mononuclear phagocyte system and neonatal Fc receptor (FcRn)-mediated recycling [139]. FcRn exports ADC to an extracellular compartment by binding in an endocytic vacuole for the recycling [140]. Therefore, antibodies have a longer half-life. On the other hand, the cytotoxic payload is metabolized in the liver and then excreted from the body through the kidneys or in the feces, and hence is affected by liver and kidney functions and alters drug–drug interactions [141].

Another challenge for ADC development is the development of drug resistance. Various mechanisms have been proposed including drug resistance to payloads, downregulation of the antigen expression level, and alteration of the intracellular trafficking pathway [142]. For example, ATP-binding cassette (ABC) transporters, such as multidrug resistance 1 (MDR1), multidrug resistance-associated protein 1 (MRP1), and breast cancer resistance protein) (BCRP), play an active role in the cellular efflux of the chemotherapeutic agents and common cytotoxic payloads such as MMAE, DM1, and ozogamicin [143,144]. Chronic exposure to T-DM1 downregulates the expression of HER2 and upregulates the MDR1 and MRP1 drug efflux pumps [145,146]. T-DXd, on the other hand, is active in HER2-positive cancer that expresses high levels of MRP2 and BCRP following resistance to T-DM1, since the payload is a poor ABC transporter substrate [147].

## 7. Future Directions

Learning from the preclinical and clinical experience of the above successful ADCs and other ADCs in development, future directions center on (1) improvement of ADC design and delivery and (2) new payloads with immunotherapy properties or radiation properties.

### 7.1. Improvement of ADC Design and Delivery

Linker chemistry, or modifying conjugation technology, is being actively studied to enhance the specificity for tumor site delivery and to reduce off-target toxicity. Hydrophilicity of the ADC molecules can be increased by introducing a PEGylated spacer, which will then improve the pharmacokinetics, tolerability, and efficacy [148,149]. Introducing a glucuronide moiety is used to protect the linker from non-specific peptidase cleavage in the systemic circulation to reduce off-target toxicity [150].

Studies are under way to modify the chemical structure of the payload. For example, the chemical structure of the DNA-crosslinking PBD-dimer was permutated to DNA-alkylating metabolites, and the indolinobenzodiazepin dimers (IGNs) led to a more efficient release of the free payload in preclinical studies [151]. Moreover, payload-binding sensitive enhancers (PBSE) have been developed which can be co-administered with ADCs, can decrease the exposure of released payload in tissues and promote clearance from plasma. [152,153].

Technologies are under way to enable more precise control of DAR and improve the stability of the payload by conjugation to more stable site [154].

Bispecific antibody technology has conveyed more prospects for ADC innovation. These novel designs employ two different paratopes aiming for both tumor specificity and lysosomal internalization. In a recent preclinical trial, a bispecific ADC targeting both HER2 and CD63 has showed better internalization and lysosomal accumulation in HER2-positive tumor cells [155].

In addition, the prospect for targeted delivery of molecules has in the recent years been extended beyond chemo-therapeutic payloads. Peptide–drug conjugates (PDCs) use polypeptide chains of about ten amino acids to target tumor antigens. PDCs offer the benefit of having a smaller molecular weight, shorter half-life, and greater tumor penetration. Currently, there are only two PDCs that have been granted FDA approval in the management of malignancies; Lu 177 dotatate, a radiolabeled somatostatin analog for the treatment of somatostatin receptor-positive gastroenteropancreatic neuroendocrine tumors (GEP-NETs) and Melflufen, used in the treatment of patients with relapsed or refractory multiple myeloma but which was later withdrawn from the market [156].

Another significant progress is the development of an ADC that employs two distinct cytotoxic agents. For example, an anti-HER2 ADC that has both MMAE and MMAF has demonstrated a more remarkable anti-tumor activity by overcoming HER2 heterogeneity in animal models [157].

### 7.2. New Payloads with Immunotherapy Property or Radiation Property

The success of ADCs has triggered the development of a class of drugs known as immune-stimulating antibody conjugates (ISACs). The mechanism of action of ISACs is through antibody-mediated targeted delivery of immune-stimulating agents that can generate an influx of pro-inflammatory cytokines, which ultimately activate dendritic cells and harness an anti-tumor T-cell response. No ISACs have so far been granted FDA approval; however, they are at a much earlier stage of development and there is hope that future clinical trials will yield promising results [158]. The concept of “immune synergy” has gained traction which focuses on combining immune checkpoint inhibitors with ADCs. This has shown promising results and is an avenue that researchers have now considered exploring further [159].

The innovative research on ADCs has also led to significant attainments regarding new payloads, specifically radioactive isotopes, such as with the success with Lu 177 dotatate. Although still currently in clinical development, the attachment of radioactive nuclei through chelating groups offers a promising avenue of treatment as it can exert a cytotoxic effect through radioactive decay. Radioactive isotopes may provide an additional advantage as these conjugates also have a diagnostic value as well [160].

## 8. Conclusions

Since the clinical application of the first ADC in 2000, Gemtuzumab ozogamicin, this class of drugs has shown rapid development and has proven to be very promising. Owing to its advantages of targeted delivery of some very potent chemotherapeutic agents, it has shown not only significant therapeutic activity in heavily treated refractory diseases, but also potent activity with prolongation of PFS and OS in earlier disease treatment. Eleven drugs are now available in the market, of which three have been approved to be used in the first line treatment setting, while all the others are undergoing randomized studies to test its activity in the early treatment arena. In solid tumors that express Her-2, ADC has shown activity in multiple tumor types including breast, esophageal, colon, and lungs, demonstrating a histology-agnostic but biomarker-dependent treatment strategy. With the innovation of bioengineering and careful clinical monitoring, the “bystander effect” can be turned into an advantage to safely and effectively treat tumors with a heterogenous antigen expression. The future of the development will see a tremendous interest in designing ADCs with improved therapeutic effect, payload-linker system with PDCs, ISACs, biospecifics, radioactive isotopes, and many other innovative designs.

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
