# Peer review of "Antibody–Drug Conjugates: A Review of Approved Drugs and Their Clinical Level of Evidence"

_cancers, 2023, doi:10.3390/cancers15153886_

Round 1
Reviewer 1 Report
Dear Authors,
Major comments:
· There are plenty of published reviews on ADCs. Everything from the first 6 pages have been written and published elsewhere many times, including numerous articles in this Cancers journal. Most paragraphs on Part 3 plainly stated the clinical response rates for different trials, very straight forward. Novelty is lacking. Could spend more effort on next generation development that might improve specificity, stability and reduce cytotoxicity of future ADCs. Could add diagrams to illustrate data or idea.
· There are arguments that non-cleavable ADCs are better than cleavable linkers, however most approved ADCs are using cleavable linkers. Which type of linker is preferred for blood or solid tumors? Any comments on the new development of linkers and next generation of inhibitors that might function superior to current approved ADC treatments?
· Can state the DAR for each ADC on table 1.
· Line 186-187. It can be dependent on the target antigen expression and function so this sentence about DAR might not be accurate.
· Most paragraphs on Part 3 plainly stated the clinical response rates in various trials. It would be useful to state the side effects in patients related to the specific antibody-drug combination and normal tissue expression level of the targets. Any drawbacks that we can learn from these clinicals for future ADC development regarding to antibody/antigen selection, patent selection or exciting drug combination?
· Please expand Part 5 or add these information for each ADC in Part 4.
· Part 6 and 7 are the most interesting sessions and the authors should expand these discussions since nearly half of the Abstract was based on ‘further exciting development’ but lacking details in the main text.
Major correction is advised.
Minor comments:
· Missing a space or two spaces after a period throughout the whole article. Please check every sentence carefully before the next submission.
· Reference before or after a period? Different throughout the whole article.
· Line 60: delete over
· Line 80: antibody dependent cellular phagocytosis (ADCP)
Author Response
Major comments:
- There are plenty of published reviews on ADCs. Everything from the first 6 pages have been written and published elsewhere many times, including numerous articles in this Cancers journal. Most paragraphs on Part 3 plainly stated the clinical response rates for different trials, very straight forward. Novelty is lacking. Could spend more effort on next generation development that might improve specificity, stability and reduce cytotoxicity of future ADCs. Could add diagrams to illustrate data or idea.
Thanks for the very insightful comment. We have included in future developments (section 7) about upcoming strategies to improve specificity, stability and reduce cytotoxicity of future ADCs. We have completely modified that section.
- There are arguments that non-cleavable ADCs are better than cleavable linkers, however most approved ADCs are using cleavable linkers. Which type of linker is preferred for blood or solid tumors? Any comments on the new development of linkers and next generation of inhibitors that might function superior to current approved ADC treatments?
We revised the description of the mechanism of action of cleaving the cleavable and non-cleavable linkers in section 2.5 extensively according to your comments. We have incorporated the above mentioned changes to explain in detail about pros and cons of each with an example. Most of the approved ADCs use cleavable linkers, and there is no preferred linker for blood versus solid tumors. Also, the new development on linkers was also added in future directions (section 7).
- Can state the DAR for each ADC on table 1.
Yes, we have added that information.
- Line 186-187. It can be dependent on the target antigen expression and function so this sentence about DAR might not be accurate.
Yes. Thanks for pointing that out. We have changed the sentence to be the following:
The ideal DAR is 2-4. A low DAR can lower the efficacy; while a high DAR may increase the drug potency, but also could negatively affect antibody structure, stability, and antigen binding, leading to faster clearance and decrease in overall clinical activity.
- Most paragraphs on Part 3 plainly stated the clinical response rates in various trials. It would be useful to state the side effects in patients related to the specific antibody-drug combination and normal tissue expression level of the targets. Any drawbacks that we can learn from these clinicals for future ADC development regarding to antibody/antigen selection, patent selection or exciting drug combination?
Taking your comments into consideration, In part 5, we have revised the manuscript and re-organized the toxicities tied to the specific antibody-drug combination, in three categories: normal tissue expression of antigen, payload side effects, and off-target organ specific side effects. We have also added a column to Table 2 to illustrate the unique and specific association of side effects to each ADC. We have also discussed improvement of ADC design, delivery and addition of new properties to ADCs in more detail in section 7.
- Please expand Part 5 or add these information for each ADC in Part 4.
Done, please see response to the above comment.
- Part 6 and 7 are the most interesting sessions and the authors should expand these discussions since nearly half of the Abstract was based on ‘further exciting development’ but lacking details in the main text.
We have incorporated changes in section 7 as advised.
Minor comments:
- Missing a space or two spaces after a period throughout the whole article. Please check every sentence carefully before the next submission.This has been revised throughout the article.
- Reference before or after a period? Different throughout the whole article. This was incoporated.
- Line 60: delete over. Deleted.
- Line 80: antibody dependent cellular phagocytosis (ADCP). This was changed.
Reviewer 2 Report
ADC research has gained significant importance in the field of cancer treatment, and it continues to generate considerable attention among researchers. As a result, I believe this review can be an interesting read for researchers interested in this area.
minor revision:
1- Please add the trade names of ADCs when they are first mentioned in the manuscript. For example, in line 51, please include the trade name of brentuximab vedotin.
2- in line 60- remove extra over after HER2
3- add Folate factor alpha (FRα) to the listed target in line 104-106
4- page 121- Please provide more information about the reasoning behind hydrozone linker cleavage ( e.g. low pH)
Author Response
ADC research has gained significant importance in the field of cancer treatment, and it continues to generate considerable attention among researchers. As a result, I believe this review can be an interesting read for researchers interested in this area.
Thank you so much for your comments. We appreciate your review.
minor revision:
1- Please add the trade names of ADCs when they are first mentioned in the manuscript. For example, in line 51, please include the trade name of brentuximab vedotin. Thanks for pointing that out. For uniformity, we included all the trade names in the table.
2- in line 60- remove extra over after HER2 : This was removed.
3- add Folate factor alpha (FRα) to the listed target in line 104-106. Added.
4- page 121- Please provide more information about the reasoning behind hydrozone linker cleavage ( e.g. low pH)
Added more information. "Hydrazone linkers are generally stable in alkaline environments, and are hydrolyzed in low pH environments, such as that in the lysosome and endosome. Hence, the cleavage of ADCs with hydrazone linkers occur predominantly in lysosome and endosome upon internalization, with occasional hydrolysis in the plasma, resulting in off-target, systemic toxicity."
Reviewer 3 Report
This educational review is well written and comprehensively covers the landscape of FDA approved ADCs both in terms of drug characteristics and in terms of the approval pathway with supporting clinical studies. There are no particular criticisms of the manuscript, which is recommended to be considered for publication in the present form.
Typos:
- page 5, line 141: The released payload will needs
- page 13, line 534: chemotherapyin
- page 14, line 561: property of the the membrane
- page 15, line 612: the first half of 202.
Author Response
This educational review is well written and comprehensively covers the landscape of FDA approved ADCs both in terms of drug characteristics and in terms of the approval pathway with supporting clinical studies. There are no particular criticisms of the manuscript, which is recommended to be considered for publication in the present form.
Thank you so much for the comments.
Typos:
- page 5, line 141: The released payload will needs.
Incorporated the change.
- page 13, line 534: chemotherapyin. Corrected (added space)
- page 14, line 561: property of the the membrane. Corrected (removed extra the).
- page 15, line 612: the first half of 202. Corrected.
Round 2
Reviewer 1 Report
Dear Authors,
Comments have been addressed satisfactorily and the paper is acceptable. Please check the formatting and grammar carefully.
Best regards